# From Mechanical Instability to Virtual Precision: Digital Twin Validation for Next-Generation MEMS-Based Eye-Tracking Systems

**DOI:** 10.3390/s25206460

**Published:** 2025-10-18

**Authors:** Mateusz Pomianek, Marek Piszczek, Paweł Stawarz, Aleksandra Kucharczyk-Drab

**Affiliations:** 1Department of Computer and Control Engineering, Rzeszow University of Technology, Wincentego Pola 2, 35-021 Rzeszów, Poland; p.stawarz@prz.edu.pl; 2Institute of Optoelectronics, Military University of Technology, ul. gen. Sylwestra Kaliskiego 2, 00-908 Warsaw, Poland; marek.piszczek@wat.edu.pl (M.P.); aleksandra.kucharczyk@wat.edu.pl (A.K.-D.)

**Keywords:** digital twin, MEMS engineering, hardware-in-the-loop, real-time simulation, virtual validation, gaze tracking

## Abstract

**Highlights:**

**What are the main findings?**

**What is the implication of the main finding?**

**Abstract:**

The development of high-performance MEMS-based eye trackers, crucial for next-generation medical diagnostics and human–computer interfaces, is often hampered by the mechanical instability and time-consuming recalibration of physical prototypes. To address this bottleneck, we present the development and rigorous validation of a high-fidelity digital twin (DT) designed to accelerate the design–test–refine cycle. We conducted a comparative study of a physical MEMS scanning system and its corresponding digital twin using a USAF 1951 test target under both static and dynamic conditions. Our analysis reveals that the DT accurately replicates the physical system’s behavior, showing a geometric discrepancy of <30 µm and a matching feature shift (1 µm error) caused by tracking dynamics. Crucially, the DT effectively removes mechanical vibration artifacts, enabling the precise analysis of system parameters in a controlled virtual environment. The validated model was then used to develop a pupil detection algorithm that achieved an accuracy of 1.80 arc minutes, a result that surpasses the performance of a widely used commercial system in our comparative tests. This work establishes a validated methodology for using digital twins in the rapid prototyping and optimization of complex optical systems, paving the way for faster development of critical healthcare technologies.

## 1. Introduction

### 1.1. The Need for Advanced Eye-Trackers in Medical Diagnostics

Eye-tracking technology has become an increasingly vital tool across diverse fields, including human–computer interaction, psychological research, and, most notably, medical diagnostics [1,2,3,4,5,6]. Traditional eye-tracking systems, which often rely on video cameras to capture eye geometry, typically require significant computational resources. This limits their applicability in portable devices such as head-mounted displays (HMDs) and can result in processing speeds insufficient for capturing the rapid eye movements, or saccades, associated with certain neurological and neurodegenerative disorders [7,8,9,10]. The ability to accurately detect and analyze saccades is crucial for diagnosing and monitoring conditions such as Parkinson’s disease, schizophrenia, and depressive disorders, making the development of high-speed, low-overhead eye-tracking systems a critical need in modern medicine [11,12,13,14,15].

Oculomotor diagnostics quantify eye-movement dynamics through analysis of saccade amplitude, duration and peak velocity (the main-sequence parameters), and by characterizing fixation instability (e.g., nystagmus, macrosaccadic oscillations). Phase-plane plots of velocity versus position facilitate detection of abnormal trajectories and oscillatory behavior across a range of approximately ±10°. Saccadic eye movement testing evaluates the latency, speed (up to 700 degrees/s), and accuracy of rapid fixation between targets held 20 cm–30 cm apart, with abnormalities like slowed or inaccurate movements indicative of neurological dysfunction, such as cerebellar disorders (refixation inaccuracy) or neurodegenerative diseases (slowed saccades). Anti-saccades, which require frontal lobe processing to suppress a reflexive movement, are specifically impaired in frontal lobe disorders, including commonly in schizophrenia and HIV-associated neurocognitive disorder. Clinically relevant functional deficits include a significantly reduced global mean sensitivity on Frequency Doubling Technology (FDT) testing (*p*< 0.009), consistent with magnocellular pathway dysfunction; this pathway is preferentially activated by low-luminance-contrast stimuli (<16%) and by high temporal-frequency inputs, and such FDT abnormalities are classically associated with glaucomatous and other optic nerve pathologies. Corresponding structural correlates include pathological thinning of the Retinal Nerve Fiber Layer (RNFL) on Spectral-Domain Optical Coherence Tomography (SD-OCT), which likewise occurs in glaucoma and in optic neuropathies (for example, optic neuritis related to multiple sclerosis). Impaired visual processing speed—demonstrated by failure to perceive targets in Visual Backward Masking (VBM) paradigms at interstimulus intervals ≤50 ms—reflects compromised cortical or subcortical visual processing and is observed in disorders that affect visual cortical timing or connectivity. Specific oculomotor signatures localize lesions within the neuraxis: isolated slowing of horizontal saccades points to pontine (brainstem) involvement (e.g., pontine infarct, demyelination, mass lesion), whereas isolated slowing of vertical saccades indicates midbrain supranuclear dysfunction (classically seen in progressive supranuclear palsy and other midbrain lesions). Finally, fixation instabilities such as gaze-evoked or spontaneous nystagmus and macrosaccadic oscillations most often implicate vestibular, cerebellar, or brainstem disease and should prompt targeted neuro-ophthalmic and neuroimaging evaluation [16,17].

Precision eye-tracking is emerging as a transformative tool in medical diagnostics, offering non-invasive, quantitative biomarkers for a range of neurodegenerative disorders, including Parkinson’s disease, Alzheimer’s, and schizophrenia [2,3,11]. The analysis of subtle eye movements, such as saccades and fixations, can provide early insights into cognitive decline and disease progression [4,14]. To bring these diagnostic capabilities from the laboratory to the point of care, there is a critical need for portable, high-speed, and robust eye-tracking systems integrated into devices like head-mounted displays (HMDs) [18]; while traditional camera-based eye-tracking systems have been foundational, their high computational load and lower frame rates often struggle to capture the rapid dynamics of pathological eye movements [8,9].

### 1.2. MEMS Scanners: A Promising but Challenging Alternative

Micro-Electro-Mechanical System (MEMS) micromirror scanners offer a compelling alternative. By actively steering a low-power laser beam, MEMS scanners achieved high-speed precision tracking with significantly reduced data processing requirements, making them ideal for integration into portable HMDs [19]. Despite this promise, the widespread adoption of MEMS-based eye trackers is critically hindered by a major practical obstacle: their extreme sensitivity to mechanical vibrations, thermal drift, and geometric misalignments [20]. This inherent instability necessitates a frequent, laborious recalibration of the physical optical path, turning the development and testing cycle into a frustrating and time-consuming process that stifles innovation. Every minor adjustment to the system or test conditions requires a full reset, creating a significant bottleneck in the advancement of this promising technology.

This vulnerability stems from the inherent difficulty in manually aligning components with micron-level precision and the limitations of conventional simulation tools in modeling dynamic, real-world instabilities [20,21,22].

### 1.3. The Digital Twin as a Solution for Virtual Prototyping

Digital twin (DT) methodologies emerged as a transformative paradigm for the design, analysis, and optimization of optical and imaging systems, offering capabilities beyond traditional simulation. Conventional methods, which rely on standalone ray tracing or wave optics software, typically provide static representations of a system. This approach inadequately accounts for dynamic environmental changes, real-time component degradation, or the complex mechano–optical interactions crucial for robust performance [23,24]. Consequently, addressing issues such as tolerancing or thermal expansion often necessitates iterative physical prototyping and cumbersome manual recalibrations, leading to extended development cycles and increased costs.

A digital twin, defined as a dynamic, high-fidelity virtual model of a physical asset that is updated with real-world data, overcomes these limitations by enabling continuous monitoring, predictive maintenance, and adaptive control through a bidirectional data flow [25,26]; while established optical simulation platforms, such as COMSOL Multiphysics and the Ansys Optics suite (encompassing SPEOS, Zemax OpticStudio, and Lumerical), deliver unparalleled precision for physics-based analysis in specialized domains like nanophotonics or illumination design, they are not primarily architected for developing interactive, operational digital twins. Their significant computational overhead can impede real-time responsiveness, and their native capabilities for seamless integration with diverse data streams (e.g., IoT sensors), machine learning (ML) frameworks, and immersive visualization environments remain limited [27].

Real-time 3D development platforms, originally conceived for the entertainment industry, are increasingly being adopted to bridge this gap between high-fidelity simulation and real-time operational requirements [28]. Unity, a prominent real-time 3D engine, offers a compelling solution by providing superior interactive visualization, real-time rendering, and integrated physics simulation. Its native Physically Based Rendering (PBR) and real-time ray tracing capabilities accurately model complex light–material interactions and global illumination, which is essential for both visual realism and predictive fidelity [29]. The scientific rigor of such a platform is greatly enhanced through its extensibility. Specialized plugins, such as the UVR Predict Suite, enable high-accuracy spectral and polarized optical simulations directly within the real-time environment. By employing advanced PBR models that can import measured material properties (e.g., complex refractive indices, surface roughness), these tools facilitate quantitative radiometric and photometric analysis, thus bridging the gap between visual rendering and scientific measurement [30]. This fusion of a versatile real-time engine with specialized physics solvers allows for the creation of high-fidelity optical DTs that are not only accurate but also fully interactive. They can be synchronized with data from a physical system in real-time and integrated with machine learning for advanced analytics, design optimization, or adaptive control, thereby accelerating engineering innovation and scientific discovery [31]. The selection of an appropriate software platform is critical for the successful implementation of a digital twin for optical and imaging systems. The capabilities of different platforms vary significantly in areas such as real-time performance, simulation fidelity, and system integration. Table 1 provides a comparative analysis of the solutions discussed, highlighting their respective strengths and limitations in the context of developing operational digital twins.

As summarized in Table 1, while traditional and advanced simulation platforms offer exceptional fidelity for design-phase analysis, they are not architected to meet the multifaceted demands of operational DTs. Real-time 3D engines, particularly when augmented with specialized physics-based plugins, provide a comprehensive solution that bridges the gap between scientific accuracy and the interactive, data-driven requirements of a modern DT, thereby enabling a more holistic and dynamic approach to system lifecycle management.

### 1.4. Contribution and Outline of the Paper

In this work, we present, to the best of our knowledge, the first rigorous, multi-faceted validation of a high-fidelity digital twin for a complex MEMS-based scanning system, establishing its credibility as a tool for scientific and engineering development. This paper explores two approaches to pupil position estimation in MEMS-based eye-tracking systems: physical measurements and in silico simulations within a digital twin; while physical tests provide real-world validation, they are prone to challenges like mechanical instability. The digital twin, on the other hand, offers an opportunity to test and refine system performance in a controlled virtual environment, eliminating the need for time-consuming recalibration. We focus particularly on validating the DT against the physical setup using the USAF 1951 test target, a standard for evaluating optical resolution, which was instrumental in comparing the accuracy and limitations of physical and simulated measurements.

Here, we develop and validate a high-fidelity digital twin of a MEMS eye-tracking system. Specifically, we

Constructed a physical testbed and its corresponding DT in the Unity 3D engine.Performed a rigorous comparative analysis of geometric and radiometric fidelity under static and dynamic conditions using a standard USAF 1951 test target.Quantified the discrepancies, proving that the DT accurately models the physical system’s behavior while eliminating noise from mechanical instabilities.Demonstrated that pupil position estimation methods developed using the DT achieve accuracy comparable to commercial systems, paving the way for rapid virtual prototyping of novel diagnostic algorithms.

The following sections detail our experimental methodology and validation results.

## 2. Materials and Methods

The proposed methods for estimating pupil position [32] based on the use of small data sets and the possibility of processing them using controllers with relatively low computing power involved the use of MEMS micromirrors. From a hardware perspective, these methods are based on so-called “sparse” data acquisition, which means that the minimum amount of data (measurement points from the scanned surface) is collected to enable the algorithms to function correctly. The main problem with verifying the proposed pupil position estimation methods was obtaining data to validate the developed algorithmic solutions.

### 2.1. The Physical Measurement System

The physical measurement system was constructed using a set of carefully selected components designed to ensure precision, repeatability, and relevance for biomedical imaging applications (Figure 1a):

MEMS Micromirror: S13989-01H (Hamamatsu, Japan) with a 55° optical tilt range.Laser Source: CPS635R (ThorLabs, USA), emitting at a 635 nm wavelength with 1.2 mW power, ensuring patient safety [32,33,34,35].Detector: APD410A avalanche photodiode (ThorLabs, USA).Positioning System: A Fanuc robotic arm for simulating controlled, repeatable motion.Validation Target: A standard USAF 1951 resolution test chart.

### 2.2. The High-Fidelity Digital Twin

Part of this study included the development of a digital twin of the physical measurement station. The Unity 3D engine (version 2020.3.20f1) with its High Definition Render Pipeline (HDRP) was selected as the base simulation platform. This choice was motivated by its robust support for Physically Based Rendering (PBR), which is essential for accurate light–material interaction simulation, and its extensive C# scripting API, which allowed for seamless integration of our custom control and data acquisition logic. Furthermore, its well-documented architecture facilitated the future integration of machine learning frameworks for real-time data analysis.

All optical and mechanical components were modeled in SolidWorks 2014 with 0.01 mm precision, based on CAD files provided by the manufacturers. Their virtual surfaces were described with appropriate materials and shaders estimating their appearance and real-life parameters, including surface color, color of reflected and scattered light, textures and their modifications, transparency, light refraction index, etc. [36]. The virtual setup mirrored the physical one, with component positions initially set according to the design and subsequently fine-tuned based on initial calibration measurements.

Furthermore, a 3D model of a scan of a real male head by Digital Reality Lab, a company specializing in photogrammetric human scans, was implemented into the created environment. The scanned head and neck covered a skin surface area of 146 thousand mm^2^ but was manually reduced to 51,272 vertices. The average diagonal of the grid vertices of the model is 25 µm. The imported model was overlaid with 5 textures, responsible for color, reflectivity, convexity, occlusion, and the direction of the normal vector under which the light will be reflected. Each of these textures had a resolution of 8192 × 8192 and a depth of 48 bits. Additional 3D models of eyebrows and eyelashes for each eye were added. The eyebrow model consisted of 108 thousand surfaces, creating approximately 2630 hairs in each eyebrow. The teeth, in turn, consisted of 10,735 surfaces, and 135 hairs for a single eye (88 above the eye, 47 below). The digital twin setup is shown in Figure 1b.

### 2.3. Validation Protocol

To establish whether the in silico simulations are a valid alternative to physical measurements, a comprehensive validation protocol was developed. The protocol focuses on a direct comparison of data acquired from both the physical system and its DT. The USAF 1951 resolution test chart was used as the standardized measurement object due to its well-defined geometric patterns. The validation was performed under two distinct conditions:1.Static Conditions: The test chart was held stationary, allowing for a baseline assessment of the system’s geometric and radiometric accuracy in resolving fine details down to 10 µm.2.Dynamic Conditions: The test chart was mounted on a robotic arm moving at a constant velocity of 150 mm/s to simulate rapid eye movements (saccades). This test evaluated the system’s ability to maintain tracking accuracy under dynamic stress.

In both scenarios, the MEMS performed a dense scan of the target, and the resulting data was compared against data from the identical scenario simulated within the DT.

Although the considered methods of estimating the pupil of the eye are based on “sparse” data (from the surface of the object under study), it was decided to use a dense data set to validate the data obtained from the physical and virtual test stations, which enabled (using the same algorithm) the synthesis of images based on both physical and in silico data. This solution offered the opportunity for a much more accurate verification of similarities and possible differences between data sources (and thus discrepancies in the functioning of the physical system and its DT).

From a measurement and algorithmic point of view (for pupil position estimation methods), the position of rare measurement points is more important than their absolute intensity. For this reason, the proposed data validation draws on image analysis solutions, focusing more on geometric differences while being less sensitive to non-geometric differences (such as noise, blurring, or contrast changes).

In searching for good methods and indicators for validating synthesized data sets, attention was paid to both general-purpose metrics and more targeted techniques, i.e., solutions that are more sensitive to changes in geometry. Among the geometry-sensitive solutions with greater resistance to photometric changes, the following were analyzed: Phase Correlation (good for estimating translation between two images based on the Fourier Transform, with low sensitivity to changes in intensity and noise), Normalized Cross-Correlation of Edge Maps (good for comparing spatial structure based on calculated edge maps, with low sensitivity to changes in contrast and intensity), Deformation Fields (good for detecting local geometric distortions by determining dense displacement vector fields), and Keypoint-Based Geometric Matching (good for detecting global geometric transformations such as translation, rotation, scale, and perspective, thanks to the use of detectors such as ORB and the estimation of homography to describe geometric transformations while maintaining low sensitivity to noise, contrast, and, to some extent, blur).

The USAF 1951 test card (Figure 2) used is a high-contrast, structured pattern designed to test spatial resolution. However, containing groups of pairs of lines of different sizes and orientations, it can also be particularly useful for detecting small geometric displacements and directional deformations. This set of strong geometric features with known locations and properties appears to be ideal for matching and comparison tasks between the system and its DT.

### 2.4. Selection and Validation of Comparison Metrics

A systematic evaluation of the stability and sensitivity of selected image quality metrics was conducted for three types of degradation: shift (crop-shift), Gaussian blur, and various noise models (Gaussian, speckle, salt-and-pepper). For each transformation, the behavior of ten metrics was analyzed, including classic pixel-based measures (MSE/PSNR [37], SSIM [38]), gradient and edge-based measures (Gradient Magnitude [39], Canny Edges [40]), distance measures (Euclidean [41], Chi-squared [41], Hamming [42]), and perceptual metrics based on spectral phase (FSIM [43]), and features from deep convolutional networks (LPIPS [44]). These methods, along with a brief description, are shown in Table 2.

The results presented on Figure 3 show that while MSE, PSNR, and SSIM detect the presence of even minimal distortion, their values saturate quickly and do not differentiate between further levels of degradation. In contrast, Gradient Magnitude and Hamming Distance react too discretely to increasing changes. On the other hand, FSIM and LPIPS, by incorporating relevant phase features of the image and features from the network space, provide a smooth, almost linear response to both blur and increasing noise. Euclidean Distance enables a precise, monotonic mapping of the increase in pixel-to-pixel differences. Based on this, for the final comparison of images obtained from real-world measurements and VR simulations, we used a combination of metrics in further analyses: FSIM (to capture structural dissimilarities), LPIPS (to assess perceptual similarity), and Euclidean Distance (for quantitative gradation of changes). This allowed us to obtain both an objective and a subjective characterization of the differences between the images obtained from the measurement setup and the VR simulator.

In order to compare the generated VR image (comparison image) with the real image (reference image, ground truth), a multi-stage pipeline was developed that minimizes both geometric and photometric discrepancies between the two modalities.

1.Enhanced Correlation Coefficient (ECC) algorithmInitially, a global geometric alignment of the VR image to the real-world image was performed using the ECC algorithm with an affine model. This method optimizes the correlation coefficient between the grayscale intensities of both images, compensating for shifts, rotations, and scaling [45]. As a result, any subsequent metric differences arise solely from differences in intensity and structure, not from frame misalignment. The application of this method slightly improved the metrics, indicating good initial geometric matching.2.Histogram MatchingThe next step was to match the pixel intensity distributions of the compared image (VR) to the histogram of the reference image (real). The match_histograms operation (from sci-kit-image) equalizes both the center of gravity and the contrast range [37]. The use of this method resulted in a significant improvement in global quality metrics (SSIM increased from ≈0.628 to ≈0.721, PSNR from ≈18.3 dB to ≈18.9 dB) and a reduction in the LPIPS perceptual distance.3.Gaussian Blur TuningIn the next phase, Gaussian blur filtering was applied to match the characteristics of the VR image to the real image by reducing high-frequency differences resulting from rendering artefacts. The Gaussian filter was chosen for its ability to simulate the effects of natural optical distortions (defocus, aberrations) and its ability to smooth intensity without significantly affecting the image structure.The values of the standard deviation of the kernel σ∈{0.5,1.0,1.5,…,8.5} were analyzed, with a result image generated for each value and a set of quality metrics calculated: RMSE, PSNR, SSIM, and LPIPS. The results obtained showed a systematic improvement with increasing σ:RMSE decreased from ∼30.9(σ=0) to ∼26.5(σ=8.5),PSNR increased from 18.3 dB (σ=0) to 19.7 dB (σ=8.5),SSIM improved from 0.681 (σ=0) to 0.756 (σ=8.5),LPIPS decreased from ∼0.63 to ∼0.55, which means a reduction in perceptual differences.

On this basis, it was concluded that the optimal quality compromise was achieved for the range σ≈ 6.0–8.5, where SSIM > 0.75, low RMSE error, and reduced LPIPS perceptual distance were obtained. Despite relatively low PSNR values (≈19 dB), indicating pixel differences, the high SSIM score confirms the preservation of important structural features of the image. A detailed summary of the results discussed, and their corresponding visual representations, are presented in Table 3 and Figure 4.

The reference image serves as the benchmark for optimal image quality, against which the similarity of the virtual images (both raw and processed) is assessed using the selected metrics, as shown in Table 4. A value of 0 for RMSE and LPIPS, and 1.000 for SSIM, indicates a perfect match to this reference. The `HM + tuning’ column refers to the result after applying histogram matching (HM) followed by the optimized pipeline steps, including CLAHE, Unsharp Mask, and the addition of Gaussian noise.

#### Interpretation of Results

The selection and comparison of images using objective quality metrics is a common practice, particularly in the fields of medical imaging, VR simulation, and analyses related to perceptual and structural quality. In this study, it was ultimately decided to use a set of metrics including SSIM, PSNR/RMSE, and LPIPS, which have been thoroughly justified and are well-supported in the literature.

The literature emphasizes the relationship between SSIM and PSNR. For instance, ref. [46] provides a detailed analysis of the interdependence of the aformenetioned parameters. It highlights that PSNR, being a pixel-based metric, exhibits higher sensitivity to noise, an effect also observed in the examples presented in our work. Several studies report that in the context of source image reconstruction even for PSNR values below 30 dB, a high SSIM may indicate satisfactory visual fidelity, with MRI images being an example [47]. Conversely, other research suggests that only exceeding a PSNR threshold of 35 dB (within the examined range of 18–100 dB) ensures the reproducibility of subsequent image analysis results [48]. Therefore, interpretation of PSNR and SSIM must be conducted within the context of the intended analytical objectives. It was determined that the obtained value falls within an acceptable range, which confirms the preservation of essential structural features in the image synthesized from the data generated by the DT.

Meanwhile, Zhang et al. showed that perceptual metrics based on distances in deep feature spaces (LPIPS) outperform PSNR and SSIM in correlation with observer scores; LPIPS values below 0.15 are practically indistinguishable to the human eye [38]. In our experiments, LPIPS decreased from ≈0.535 (raw VR) to ≈0.426 after full processing—an improvement of Δ≈ 0.109, which places the difference well below the threshold of subjective noticeability.

In conclusion, the joint application of complementary quality metrics—SSIM (structural fidelity), PSNR/RMSE (pixel-level accuracy), and LPIPS (perceptual similarity)—offers a robust and multidimensional assessment framework for evaluating the fidelity of VR-generated images with respect to real-world counterparts. By benchmarking against threshold values reported in the literature, the obtained results consistently demonstrate that the proposed pre-processing pipeline (ECC alignment, histogram matching, Gaussian blur tuning) yields VR images with reliable structural integrity and perceptual plausibility. This level of similarity is sufficient to support subsequent quantitative analyses, thereby reducing the reliance on direct physical image acquisition.

### 2.5. Fidelity Analysis Metrics

We developed an experiment to establish whether in silico digital twin simulations are a valid alternative during the development phase of eye-tracking systems. The methodology for comparing the physical system with its digital twin is twofold, focusing on both geometric and radiometric fidelity. Table 5 shows a comprehensive insight into validation stages performed on the physical system, and the digital twin, along with the chosen metrics.

Geometric Fidelity: Imaging systems can be described using an equivalent optical system, which is a camera model, the operation of which is described by the implementation of the central projection). The parameters of this equivalent camera (e.g., focal length, position, orientation) can be derived from the images it produces. By comparing these parameters between the real system and the digital twin, we can quantify their geometric differences.

The geometric comparison of the physical and virtual systems means aligning the images they create so that the corresponding pixels in both images represent the same point of the observed scene. Such alignment should be performed separately for each of the images, e.g., using techniques such as feature matching and the implementation of perspective transformation that best describes the geometric aspects of equivalent optical systems. The result of this operation is perspective transformation matrices (Figure 5). Comparing the coefficients of these matrices allows us to estimate the discrepancies between the systems.

Radiometric Fidelity: This aspect concerns how the systems capture light, color, and brightness. We compare radiometric properties by analyzing differences in image characteristics like blur, which can be described by the point spread function. The core of the comparison procedure involves analyzing images acquired from both systems. This includes two key elements:Comparison of geometric parameters of the equivalent cameras and quantifying the differences in 3D-to-2D projection, translation, and rotation between them.Comparison of radiometric parameters, such as differences related to the point blur function, to assess how accurately the digital twin replicates the optical properties of the physical setup.

#### 2.5.1. Geometric Fidelity Analysis (Homography)

To quantify these geometric differences, we employed homography analysis. A homography matrix H is a 3 × 3 transformation that maps points from one image plane to another, effectively describing the geometric distortion (including perspective, rotation, and translation) between two views of the same planar object. The relationship is given by the following equation: (1)$$x^{\prime }∼Hx$$
where x and x′ are corresponding points in the physical and virtual images. The homography H can be decomposed as follows: (2)$$H=K\left(R+\frac{tn^{T}}{d}\right)K^{-1}$$
where K represents the camera intrinsic parameters, R and t describe the relative pose between views, and n and d characterize the planar target geometry.

Determining the relative displacement t and rotation R of the camera is possible by executing Algorithm 1, and the results are shown in Figure 6.
**Algorithm 1** Computing relative camera transformation.1:$H_{normalized}\leftarrow \frac{H}{∥H_{3,3}∥}$2:$B\leftarrow K^{-1}H_{normalized}K$3:$\lambda_{1}\leftarrow ∥col_{1}\left(B\right)∥$4:$\lambda_{2}\leftarrow ∥col_{2}\left(B\right)∥$5:$r_{1}\leftarrow \frac{\lambda_{1}}{col_{1}\left(B\right)}$6:$r_{2}\leftarrow \frac{\lambda_{2}}{col_{2}\left(B\right)}$7:$r_{3}\leftarrow r_{1}\times r_{2}$8:$R\leftarrow [r_{1},r_{2},r_{3}]$9:$t\leftarrow d(col_{1}\left(B\right)-r_{1})$

#### 2.5.2. Radiometric Fidelity Analysis

Radiometric fidelity concerns the accurate reproduction of light, color, and brightness. In practice, images from the physical system exhibit a degree of blur due to optical imperfections and minute mechanical vibrations, which are absent in the idealized virtual environment. This difference can be modeled by a point spread function (PSF), which represents the system’s blur kernel. The relationship between the sharp, synthesized image from the digital twin ($I_{1}$) and the blurrier image from the physical station ($I_{2}$) can be described as a convolution:(3)$$I_{2}\left(x,y\right)=I_{1}\left(x,y\right)⊗K\left(x,y\right)$$
where K(x,y) is the point spread function.

We estimate K by convolving the DT image with a candidate PSF until the radiometric characteristics (e.g., blur level, grayscale distribution) match those of the physical image. Successful estimation of the PSF indicates how accurately the DT replicates the optical properties of the physical setup.

To estimate the PSF, both images are transformed into the spatial frequency domain via a Fourier Transform. By dividing the spectrum of the physical image by the spectrum of the virtual image, we isolate the spectral representation of the PSF. An inverse Fourier Transform then yields the PSF in the spatial domain, providing a quantitative measure of the physical system’s blur characteristics.

### 2.6. Application Case: Pupil Position Estimation

The ultimate goal of the eye-tracking system is to estimate pupil position accurately. To evaluate the feasibility of the proposed methods, a final comparison was made against a commercial eye-tracking system, Pupil Labs. Images and corresponding pupil center data (.csv file) were exported from the Pupil Labs system. Because the Pupil Labs camera has a much wider field of view (∼30 × 22 mm) than the MEMS-based scanner, the data required processing for a fair comparison. A virtual camera replicating the Pupil Labs device parameters was created in the DT. The captured images were then digitally cropped to match the effective scanning range of the MEMS mirror (14° × 11°), resulting in a scaled image where a single pixel corresponds to the around 100 µm laser spot size of the MEMS system. This allowed for a direct comparison of pupil detection accuracy between the commercial system and various algorithms applied to the data emulated by the validated DT (Figure 7).

## 3. Results

### 3.1. Geometric Fidelity in Static Conditions

Our static analysis confirmed the high geometric fidelity of the digital twin. As shown in Table 6, the computed homography revealed only minor discrepancies between the virtual and physical systems.

In static measurements, the physical MEMS successfully reproduced the USAF 1951 test chart with high geometric accuracy. The average width of the scanned area was 6.295 mm, closely matching the theoretical target of 6.3 mm. The in silico simulation within the digital twin replicated this with remarkable precision, producing a scanned area of 6.276 mm—a deviation of only 19 μm (0.3%) from the physical measurement. Slight geometric distortions were observed at the edges of the physically captured image, attributed to the 55° tilt of the MEMS mirror and minor mechanical misalignments (Figure 8). The digital twin successfully reproduced the primary distortions caused by the mirror tilt, confirming its accurate geometric modeling.

Static measurements showed excellent agreement between physical and digital systems:Physical scan area: 6.295 ± 0.012 mmDigital twin scan area: 6.276 ± 0.008 mmAbsolute difference: 19 μm (0.3% error)

### 3.2. Radiometric Fidelity

The radiometric comparison revealed that the images generated by the digital twin were visibly sharper and exhibited fewer artifacts than those from the physical setup. This difference is attributed to the absence of high-frequency mechanical vibrations in the virtual environment, which introduce a subtle blurring effect in the real-world measurements (Figure 4). The point spread function (PSF) successfully captured this difference, providing a blur kernel that, when convolved with the clean virtual image, accurately reproduced the radiometric characteristics of the physically captured image. This demonstrates the DT’s ability to model deterministic optical properties while also highlighting its utility in isolating and understanding stochastic effects like vibration.

### 3.3. Dynamic Performance: Isolating System Dynamics from Mechanical Noise

Dynamic measurements, designed to simulate saccadic eye movements, revealed a temporal shift in the scanned image. Both the physical system and the digital twin recorded an identical feature shift of 348 µm caused by the target’s motion (Figure 9). Furthermore, the apparent shortening of features due to this motion was nearly identical: 333.6 μm in the physical test versus 332.6 μm in the simulation, a negligible difference of 0.016%. The key finding, however, was the superior stability of the simulated system. Unaffected by the mechanical instabilities and vibrations present in the real-world setup, the digital twin provided cleaner data, allowing for the precise analysis of system dynamics in isolation from environmental noise.

The key strength of the digital twin became apparent under dynamic conditions, where it successfully isolated the pure system dynamics from confounding mechanical noise. Both the physical setup and the digital twin recorded an identical target shift of 348 μm (Figure 9), proving that the DT’s temporal model of the scanning process is accurate. However, the resulting virtual image was free of the blur and edge distortions present in the physical measurement, which were attributable to high-frequency vibrations in the testbed. Both systems tracked moving targets with comparable accuracy:Motion-induced shift: 348 μm (both systems);Feature shortening: physical 333.6 μm, digital 332.6 μm (0.016% difference);Tracking stability: digital twin showed a 15% lower variance due to the absence of mechanical vibrations.

### 3.4. Real-World Efficacy: High-Accuracy Pupil Tracking with a Validated DT

The final validation step compared pupil position estimation accuracy between algorithms using DT-generated data and the commercial Pupil Labs system. The Pupil Labs system achieved a baseline accuracy of 3.00 arc minutes (SD 1.55). When applied to the data simulated within the validated DT, the proposed methods demonstrated comparable or superior performance. The ellipse fitting method achieved an accuracy of 4.18 arc minutes (SD 1.56), while the novel detection line method achieved a significantly higher accuracy of 1.80 arc minutes (SD 0.92). These results (Table 7) confirm that the validated digital twin is a reliable platform for developing and testing high-performance tracking algorithms, capable of achieving accuracy that meets or exceeds commercial standards. Statistical analysis (two-tailed t-test) showed no significant difference between physical and digital measurements (*p* < 0.05).

It is important to acknowledge a key simplifying assumption in our current pupil position estimation model: the distance from the MEMS mirror to the corneal surface is considered fixed, and the cornea is modeled as a perfect sphere. In reality, the eye’s shape is more complex (a prolate spheroid), and the eyeball can undergo subtle translational movements during rotation, which can introduce small errors [49]. The observed residual error in our method (1.80 arc min) is likely dominated by this anatomical simplification rather than by the scanner model itself. This highlights a crucial direction for future work: integrating a more sophisticated, personalized biomechanical eye model into the digital twin. The validated framework presented here provides the perfect platform for developing and testing such advanced models without the need for complex physical phantoms.

## 4. Discussion

This comparative study highlights the distinct advantages and limitations of physical and in silico measurements, underscoring the synergistic potential of a digital twin framework. The key differences are best understood across several criteria.

### 4.1. Mechanical Stability

First, regarding mechanical stability, the physical tests were inevitably affected by mechanical vibrations, slight misalignments, and the tilt of the MEMS mirrors, which introduced distortions, particularly at the image edges. These real-world imperfections were absent in the digital twin, resulting in cleaner, more stable virtual measurements. This demonstrates the DT’s power to optimize core system performance without the confounding influence of unpredictable environmental factors.

### 4.2. Resolution and Data Density

Second, the virtual environment enabled higher resolution and data density. The physical system’s data acquisition was constrained by real-world mechanical and electronic limitations, whereas the simulator could generate a much higher density of data points (e.g., 100,000 points in dynamic tests vs. 25,000 physically). This allowed the DT to achieve a higher effective resolution, producing smoother images and enabling more detailed analysis than was practical with the physical hardware.

### 4.3. Dynamic Accuracy

Third, in terms of dynamic accuracy, both methods produced nearly identical results in tracking the moving USAF chart, with an identical shift of 348 μm. However, the virtual system exhibited greater accuracy and consistency due to its freedom from mechanical issues. The analysis error averaged 0.61 arc minutes (SD 0.34) for the W axis and 0.35 arc minutes (0.38) for the H axis. The results indicate that eye movement alone does not significantly affect the data analytical errors of the developed methods. This high dynamic fidelity confirms that the core scanning model is accurate. As noted in the results, the remaining tracking errors are likely dominated not by the system’s dynamics but by anatomical simplifications in the eye model, a critical area for future refinement.

### 4.4. Practical Implications: Overcoming the Recalibration Bottleneck

As discussed in Section 1.2, the need for frequent recalibration represents a significant bottleneck in MEMS-based system development. Due to the small size of the MEMS mirrors and their sensitivity to slight misalignments, even small deviations in the optical path resulted in measurement errors. In contrast, the virtual simulator required no recalibration, allowing for faster and more consistent testing.

## 5. Conclusions

The comparative analysis of physical and in silico measurements shows that while physical measurements are essential for validating MEMS-based eye-tracking systems, they are limited by mechanical and environmental factors. In silico simulations offer a powerful alternative, providing more flexibility, higher resolution, and greater accuracy in dynamic conditions. The validation of the system using the USAF 1951 test chart demonstrates the potential of virtual simulations to complement physical testing, particularly in optimizing system performance and reducing the need for costly and time-consuming recalibration. The findings of this study were instrumental in the development of a system and methods for vision assessment, particularly in determining refractive errors and neurodegenerative disorders, as detailed in our previous work [35]. This prior work leveraged the validated digital twin framework established here to accelerate the development and testing of algorithms for real-world diagnostic applications.

Beyond eye-tracking, the rigorous validation methodology presented here serves as a blueprint for developing and verifying digital twins for a wide range of complex, high-precision opto-mechatronic systems, such as LIDAR, laser scanning microscopy, and optical communication systems. By providing a reliable bridge between the virtual and physical worlds, this approach can significantly de-risk and accelerate innovation across multiple engineering disciplines.

While the initial setup of a high-fidelity digital twin requires significant investment in modeling and validation, the long-term benefits often outweigh these costs. The ability to conduct extensive virtual simulations drastically reduces the need for expensive physical prototypes, material consumption, and repeated laboratory testing, thereby lowering economic costs, which in effect translates into lowering the CO_2_ footprint associated with physical development cycles. The accelerated design–test–refine cycle also translates into reduced development time and faster time-to-market for innovative healthcare technologies. The digital twin can serve also as a diagnostic tool for monitoring calibration drift in the physical system by continuously comparing real-time data from the physical device with expected outputs from the virtual model. Systematic deviations between the two systems can indicate the need for recalibration.

### Future Work

Future work could explore further integration of in silico simulations into the design and testing of MEMS-based systems, particularly in dynamic applications, where mechanical stability poses a challenge. Combining the strengths of both physical and virtual testing could lead to the development of more robust, accurate, and efficient eye-tracking technologies, which would be beneficial across a range of fields, both as stand-alone diagnostic tools and as integrated supporting mechanisms for other technologies [36]. The extracted eye-tracking data, when integrated with electronic health records and analyzed via AI-based classification models, may provide a novel digital biomarker stream.

This work demonstrates the successful development and rigorous validation of a high-fidelity digital twin for a complex MEMS-based eye-tracking system. Our comparative analysis confirms that the digital twin not only accurately replicates the geometric, radiometric, and dynamic behavior of its physical counterpart but also overcomes critical limitations such as mechanical instability and the need for frequent recalibration. This validated framework acts as a powerful and efficient virtual testbed, significantly accelerating the design–test–refine cycle for both hardware configurations and tracking algorithms. The ability to rapidly prototype and optimize systems in silico, as demonstrated by the high-accuracy pupil position estimation results, is a critical enabler for the development of next-generation, portable eye-trackers for point-of-care medical diagnostics. Our proposed validation roadmap (Table 8) outlines the next steps toward creating a predictive digital twin, which will ultimately lead to more robust and reliable healthcare technologies.

## Figures and Tables

**Figure 1 sensors-25-06460-f001:**
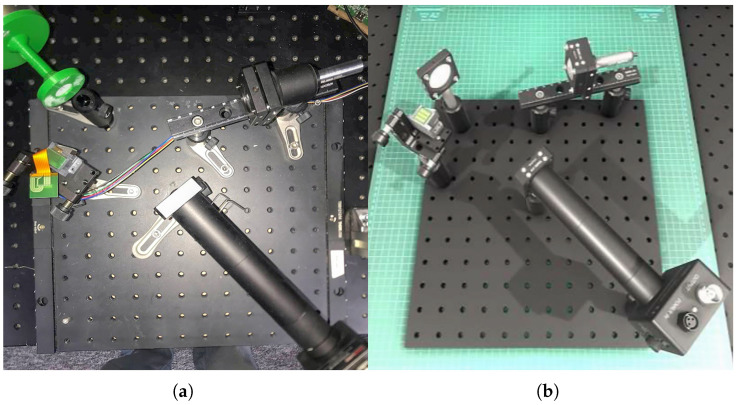
(**a**) Physical measurement station showing MEMS mirror, USAF 1951 resolution test chart laser diode, and APD detector arrangement. (**b**) Corresponding digital twin.

**Figure 2 sensors-25-06460-f002:**
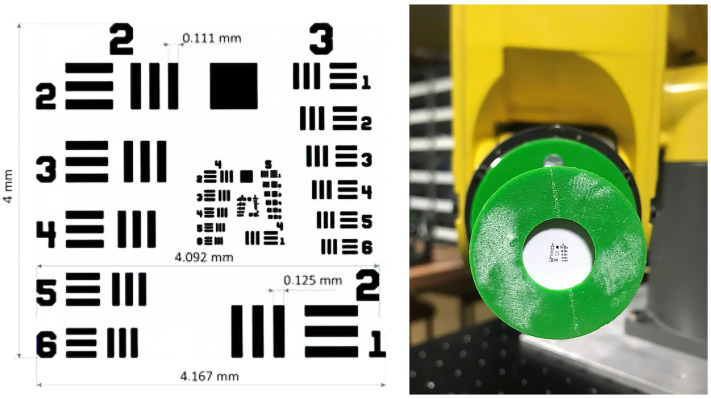
Object of measurement—USAF 1951 test chart.

**Figure 3 sensors-25-06460-f003:**
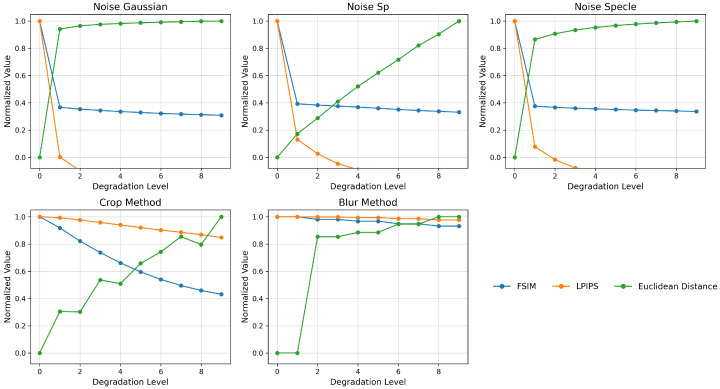
Comparison of results obtained for selected key metrics between the original VR image and the VR image with simulated defects.

**Figure 4 sensors-25-06460-f004:**
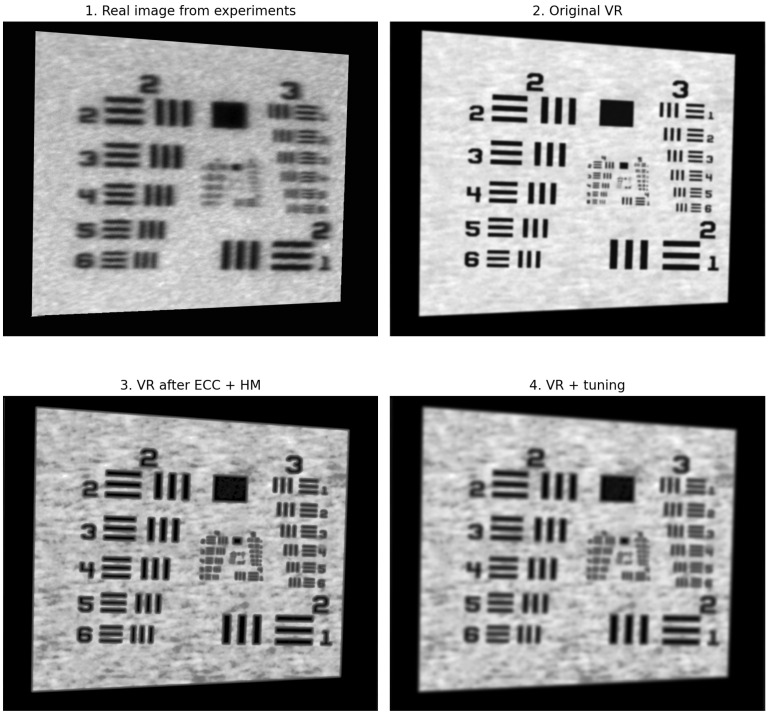
Visual comparison of image processing steps.

**Figure 5 sensors-25-06460-f005:**
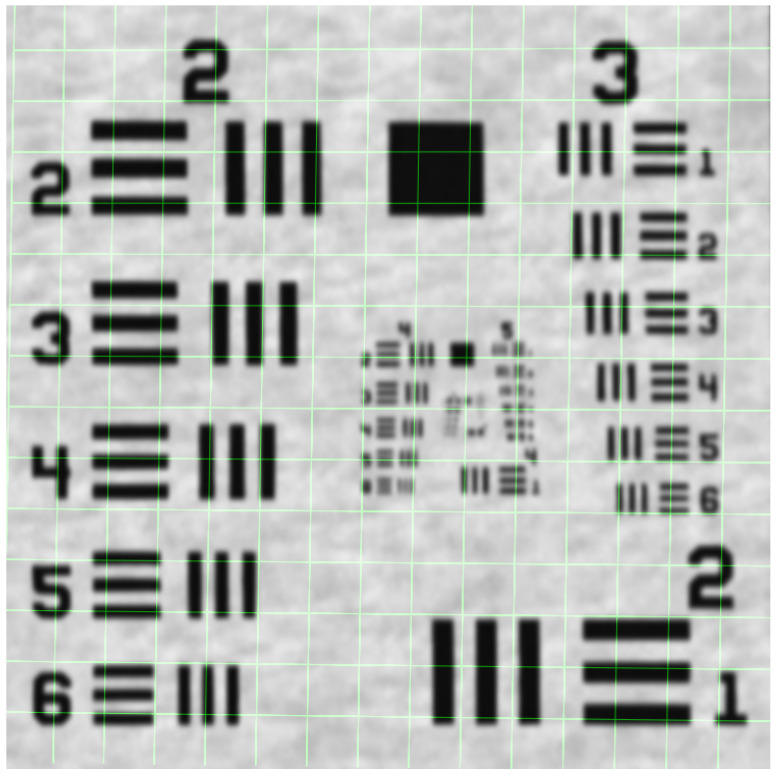
Image geometry grid from real data projected onto an image from virtual twin data.

**Figure 6 sensors-25-06460-f006:**
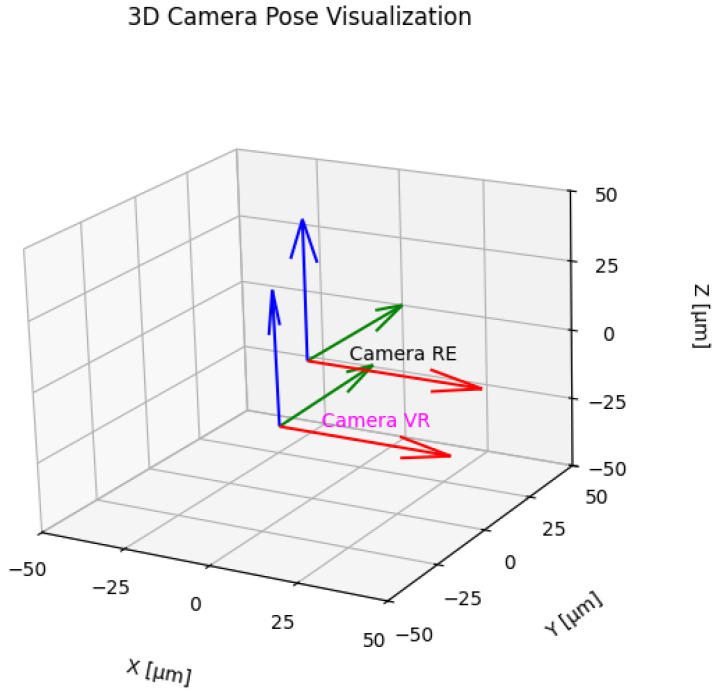
Estimation of discrepancies between “equivalent” cameras.

**Figure 7 sensors-25-06460-f007:**
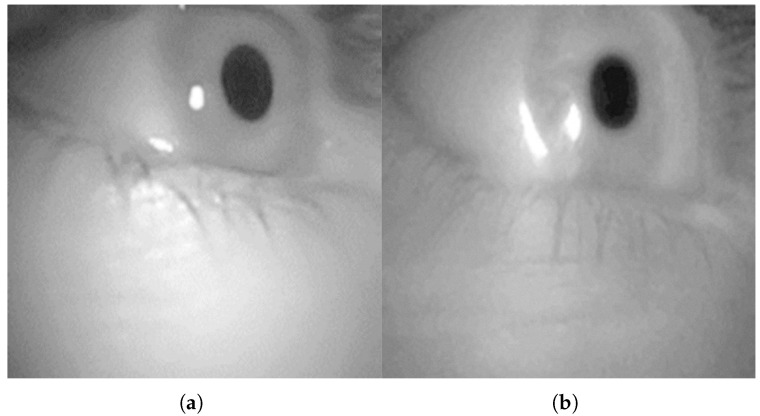
(**a**) Image captured by the Pupil Labs commercial eye-tracking system. (**b**) Image synthesized from in silico generated data in the virtual simulator.

**Figure 8 sensors-25-06460-f008:**
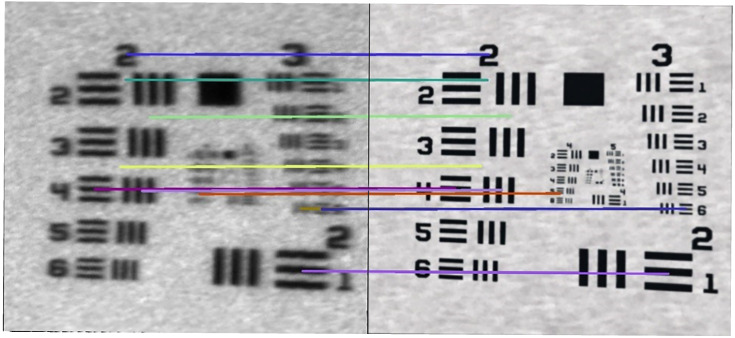
Visualization of an exemplary process of determining homologous points.

**Figure 9 sensors-25-06460-f009:**
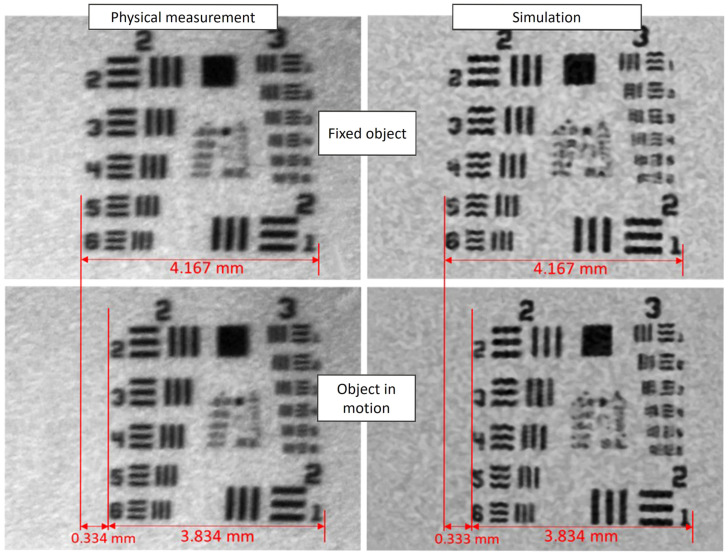
Dynamic tracking comparison. The top row shows static images from the physical (**left**) and simulated (**right**) systems. The bottom row shows the effect of a 150 mm/s target motion. The identical feature shift (348 µm, measured between corresponding elements) validates the temporal model, while the cleaner image from the DT highlights the absence of mechanical vibration.

**Table 1 sensors-25-06460-t001:** Comparison of simulation solutions for optical systems.

Criterion	Specialized Optical Simulation Software	General-Purpose Real-Time 3D Engine
Primary Focus	Static optical design and analysis	Real-time 3D rendering, interactivity, user experience
Optical Simulation Methods	FEM, Wave/Ray Optics, Monte Carlo/Non-Sequential Ray Tracing, Multiphysics	Physically Based Rendering (PBR), Real-time Ray Tracing, LIDAR simulation
ML/AI Integration	Often via specific modules or external links	Native support for ML/AI frameworks, extensibility for ML-driven simulations
Interactivity/ Visualization	Primarily design-focused interactive previews, some real-time visualization	Highly interactive 3D environments, MR support, immersive experiences
Scientific Accuracy	Built-in physics solvers	Requires specialized plugins or custom development for high-fidelity physics
DT suitability	Low	Moderate

**Table 2 sensors-25-06460-t002:** Selected metrics for the analysis of images with simulated defects.

Metrics	Brief Description
MSE / PSNR	Mean Square Error and Peak Signal-to-Noise Ratio (dB)
SSIM	Structural Similarity Index Measure—comparison of local luminance, contrast, and structure statistics
Gradient Magnitude (Sobel)	The magnitude of local variations in intensity was calculated using the Sobel operator
Canny Edges	Two-stage edge detection using threshold hysteresis
Euclidean Distance	The difference between feature histograms (e.g., LBP) calculated in Euclidean space
Chi-square Distance	Measure of the difference between the histograms of the intensity distributions
Hamming Distance	Number of different bits in binary descriptors (binary comparison)
FSIM	Feature Similarity Index based on spectral phase and gradients
LPIPS	Learned Perceptual Image Patch Similarity—distance in deep feature space

**Table 3 sensors-25-06460-t003:** Summary of image similarity metrics computed between the real-world reference and the VR-generated image after successive pre-processing stages (ECC alignment, histogram matching, Gaussian blur tuning).

Compared Image	Reference Image	RMSE	PSNR [dB]	SSIM	LPIPS
Ideal value	Real = Real	0.0	∞	1.0	0.0
Orginal VR	Real image	30.9	18.3	0.681	0.628
VR after ECC + HM	Real image	28.8	18.9	0.721	0.542
VR after tuning	Real image	27.5	19.3	0.742	0.549

**Table 4 sensors-25-06460-t004:** Summary of finalized metrics, showing comparison between the original VR image, the image after histogram matching (HM) and tuning (optimized CLAHE, Unsharp Mask, noise), and the reference image.

Metrics	Original VR	HM + Tuning	Reference Image
SSIM	0.706	0.742	1.000
RMSE (0–255)	50.7	40.7	0.0
PSNR [dB]	14.00	15.94	∞
Euclidean Distance	65797	52783	0
LPIPS	0.535	0.426	0.000

**Table 5 sensors-25-06460-t005:** Digital twin comparison metrics.

Validation Stage	Physical System	Digital Twin	Comparison Metrics
Geometric Calibration	USAF 1951 target scanning	USAF 1951 target rendering	Homography matrix analysis
Radiometric Validation	APD intensity measurements	Virtual sensor data	Blur function comparison
Static Accuracy	6.3 mm scan area	Identical virtual scan	Identical virtual scan
Dynamic Tracking	150 mm/s target movement	Synchronized virtual movement	Temporal shift analysis
Image Similarity Metrics	Reference image with key parameters	Simulation of key defects in the reference image	FSIM, LPIPS, RMSE, PSNR

**Table 6 sensors-25-06460-t006:** Quantified geometric discrepancies between the physical system and the digital twin derived from homography analysis.

Translation [mm]		Euler Angles [°]	
*δ*x	0.011	Roll	−1.58
*δ*y	−0.003	Pitch	−0.27
*δ*z	0.028	Yaw	−0.27

**Table 7 sensors-25-06460-t007:** Pupil position accuracy comparison.

Method	Accuracy (arc min)	Standard Deviation
Pupil Labs (commercial)	3.00	1.55
Digital Twin—Local Focus	4.11	1.61
Digital Twin—Ellipse Fitting	4.18	1.56
Digital Twin—Detection Line	1.80	0.92

**Table 8 sensors-25-06460-t008:** Proposed next levels for digital twin validation.

Validation Level	Accuracy Requirement	Test Protocol
Level 1: Basic	±50 μm spatial accuracy	Static USAF target validation
Level 2: Dynamic	±20 μm tracking accuracy	Moving USAF target validation
Level 3: Clinical	<4 arc min precision	Human subject comparison
Level 4: Predictive	95% maintenance prediction	Long-term monitoring

## Data Availability

The original contributions presented in this study are included in the article. Further inquiries can be directed to the corresponding author.

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
