# Peer review of "From Mechanical Instability to Virtual Precision: Digital Twin Validation for Next-Generation MEMS-Based Eye-Tracking Systems"

_sensors, 2025, doi:10.3390/s25206460_

Round 1

Reviewer 1 Report

Comments and Suggestions for Authors

This manuscript proposes and validates a high-fidelity digital twin (DT) model tailored for next-generation MEMS-based eye-tracking systems. Through systematic comparative experiments between the physical system and its virtual counterpart, the study demonstrates a high degree of consistency in terms of structural accuracy, radiometric fidelity, and dynamic response. Furthermore, the digital twin is employed to validate pupil localization algorithms, achieving performance that surpasses that of existing commercial systems. Overall, the work is novel, methodologically rigorous, and offers valuable insights for MEMS system engineering and the application of digital twin technology in biomedical optics. Here are two suggestions:

1. Section 2.4 is somewhat lengthy; it is recommended that it be shortened. 

2. It is recommended that further information be provided on the outlook for future work.

Author Response

We are very grateful for the kind words regarding our manuscript, and have tried our best to improve it even further by implementing all suggestions.

Comments 1: Section 2.4 is somewhat lengthy; it is recommended that it be shortened. 
Response 1: We understand the suggestion. Since so much has been done during the research process, we had problems with whether to include some things. In the reviewed manuscript, most of section 2.4 has been rewritten in an attempt to condense the information contained within.

Comments 2: It is recommended that further information be provided on the outlook for future work.
Response 2: We have expanded the "Conclusion" section in multiple ways. First of all, as a different reviewer pointed out, we added more information regarding the results of the experiments. Secondly, we provided additional examples of research that can be performed in the future, and is connected to the subject of our manuscript.

Reviewer 2 Report

Comments and Suggestions for Authors

The authors describe the usage of a digital twin to better develop and verify a high-performance MEMS eye tracker. The method of using Unity to set up the digital twin is quite interesting and new to me. Overall the paper is good to read and I recomment publication as is.

Author Response

We are sincerely grateful for this kind review, and although, following other reviews, some revisions were made to the manuscript, we hope it has only improved the overall quality. We have chosen Unity for several reasons, but most notably because it offers easy access to high-definition graphics, combined with ease of programming.

Reviewer 3 Report

Comments and Suggestions for Authors

This papers focuses on the comparison of Digital Twin with in-silico measurements and physical MEMS-based eye-tracking systems, using a standard test chart (USAF 1951) as a test bench. It demonstrates the potential and limits of virtual simulations, that could be used for improving and optimizing the overall performance, while limiting the need for MEMS and system recalibration.
One of the main interest of the Digitial Twin is its freedom from mechanical issues and vibrations.
The findings of this study have already been used in the development of system and methods to determine refractive errors and neurodegenerative disorders.

Could the digital twin be used for determining when physical calibration is needed ? and to add a script or work on experimental live data to add corrections due to physical limitations and uncertainties such as vibrations ?

The authors could have discussed in more details the "cost" (economic, development time, CO2 footprint) of the implementation of  a Digital Twin.

In chapter 1. Introduction, it could be interesting to add few quantitative values, such as the rapid eye movements (in order of 200mm/s).

In Figure 5, units on the X and Y axis could be in µm instead of mm, that seems to be more appropriate.

Author Response

Comments 1: Could the digital twin be used for determining when physical calibration is needed ? and to add a script or work on experimental live data to add corrections due to physical limitations and uncertainties such as vibrations ?
Response 1: Yes, the digital twin can serve as a diagnostic tool for monitoring drift in the physical system, mainly by continuously comparing real-time data from the physical device with expected outputs from the virtual model. We have added a brief mention of the subject to the "Conclusion" section of the paper.

Comments 2: The authors could have discussed in more details the "cost" (economic, development time, CO2 footprint) of the implementation of  a Digital Twin.
Response 2: We have considered this suggestion really interesting, and decided to further expand the "Conclusion" section, by adding all information that was available to us regarding the cost of the implementation. We did not research the CO2 footprint, but since the base tools are general use, the footprint should be minimal.

Comments 3: In chapter 1. Introduction, it could be interesting to add few quantitative values, such as the rapid eye movements (in order of 200mm/s).
Response 3: In the original manuscript we wanted to focus on the development of the Digitial Twin, and as such, we only provided an outline of interconnected subjects. However, we do agree quantitative values regarding saccades would improve the overall quality of the manuscript, and thus have expanded chapter 1 to include them.

Comments 4: In Figure 5, units on the X and Y axis could be in µm instead of mm, that seems to be more appropriate.
Response 4: Figure 5 has been updated, as per the suggestion, which we agree was a oversight on our part.

Reviewer 4 Report

Comments and Suggestions for Authors
  1. In Table 3, please explicitly state what the abbreviation “HM” stands for. Is it histogram matching? Also, what does the “tuning” process involve?
  2. In Table 3, what is considered the reference image? How does it differ from the “Original VR” and “HM+tuning,” and what is its specific role in the comparison?
  3. Regarding Table 3, it may also help readers better understand the effectiveness of the metrics if the authors could provide some visual examples of the “Original VR,” “HM+tuning,” and the “reference image.”
  4. In the conclusion, the authors state that “The findings of this study were used in the development of [33].” It would be helpful if the authors could include a brief description of the referenced work in such cases, rather than citing only the reference number.

Author Response

Comments 1: In Table 3, please explicitly state what the abbreviation “HM” stands for. Is it histogram matching? Also, what does the “tuning” process involve?
Response 1: We did indeed make the mistake of including the abbreviation "HM" without explaining it. As suggested in the review, "HM" stands for "histogram matching". We have added an explanation in the table description and below the table. Moreover, we included "HM" in the abbreviations list. We have also added a description of the tuning process to make things clearer.

Comments 2: In Table 3, what is considered the reference image? How does it differ from the “Original VR” and “HM+tuning,” and what is its specific role in the comparison?
Response 2: These are very valid questions. The reference image is essentially the image obtained by the real system, which serves as the benchmark for optimal image quality, against which the similarity of the virtual images (both raw and processed) is assessed using the selected metrics. We hope the expanded description in chapter 2.4 proves itself helpful in explaining the topic more clearly.

Comments 3: Regarding Table 3, it may also help readers better understand the effectiveness of the metrics if the authors could provide some visual examples of the “Original VR,” “HM+tuning,” and the “reference image.”
Response 3: We agree with the suggestion that the manuscript would benefit from including a visual comparison. The revised manuscript contains a visual comparison (Figure 8). 

Comments 4: In the conclusion, the authors state that “The findings of this study were used in the development of [33].” It would be helpful if the authors could include a brief description of the referenced work in such cases, rather than citing only the reference number.
Response 4: Briefly speaking, reference [33] describes methods along with a system created by applying them, which is used for vision assessment, particularly in determining refractive errors and neurodegenerative disorders. We do understand that access to [33] is limited, and included the description in the revised version of the manuscript.

Round 2

Reviewer 4 Report

Comments and Suggestions for Authors

The authors have addressed the comments and provided clear explanations of key terminology, illustrative image examples, and concise elaborations on the references cited at the appropriate locations in the manuscript. These additions will significantly enhance the accessibility and comprehension of the work for readers. I have no further questions.